# Densification, Microstructure and Anisotropic Corrosion Behavior of Al-Mg-Mn-Sc-Er-Zr Alloy Processed by Selective Laser Melting

Jinglin Shi [1,2], Qiang Hu [1,*], Xinming Zhao [3,*], Jiahao Liu [2], Jiacheng Zhou [2], Weichen Xu [2] and Yaolong Chen [2]

1    GRINM Group, Industrial Research Institute for Metal Powder Material, Beijing 101407, China
2    General Research Institute for Nonferrous Metals, Beijing 100088, China
3    GRINM Additive Manufacturing Technology Co., Ltd., Beijing 101407, China
*    Correspondence: hqgrinm@163.com (Q.H.); xinming_zhao@126.com (X.Z.)

**Abstract:** High-performance additives manufactured by Al alloys provide significant potential for lightweight applications and have attracted much attention nowadays. However, there is no research on Sc, Er and Zr microalloyed Al alloys, especially concerning corrosion behavior. Herein, crack-free and dense Al-Mg-Mn-Sc-Er-Zr alloys were processed by selective laser melting (SLM). After optimizing the process parameters of SLM, the anisotropic corrosion behavior of the sample (volume energy density of 127.95 J·mm$^{-3}$) was investigated by intergranular corrosion (IGC) and electrochemical measurements. The results showed that the XY plane of the as-built sample is less prone to IGC, and it also has a higher open circuit potential value of $-901.54$ mV, a higher polarization resistance of $2.999 \times 10^4$ Ω·cm$^2$, a lower corrosion current of 2.512 µA·cm$^{-2}$ as well as passive film with superior corrosion resistance compared to the XZ plane. According to our findings, the XY plane has superior corrosion resistance compared to the XZ plane because it has fewer primary phases of Al$_3$(Sc, Er, Zr) and Al$_2$MgO$_4$, which can induce localized corrosion. Additionally, a higher fraction of low-angle grain boundaries (LAGBs) and a stronger (001) texture index along the building direction are also associated with better corrosion resistance of the XY plane.

**Keywords:** selective laser melting; Al-Mg-Mn-Sc-Er-Zr alloy; densification; microstructure; anisotropic corrosion behavior





## 1. Introduction

The aerospace, marine and automotive industries are driven by the requirements of energy conservation, emission reduction and cost savings, so lightweight applications have gained widespread attention [1]. Topological optimization design can be used to reduce the weight of components in service, but it is time-consuming, costly and wasteful to produce material components with complex structures using conventional production technology [2]. Additive manufacturing (AM) is able to fulfill the above-mentioned requirement, which involves multi-disciplines such as materials, machinery, computer and numerical control [3,4]. The basic process of AM is to divide the three-dimensional (3D) structure model of a part into equal-thickness cross-sections and progressively add materials in layers to build the part [3]. On the other hand, as a lightweight material, aluminium (Al) alloy has vast applications for weight reduction as its desirable specific strength and rigidity, good machinability and formability, extraordinary corrosion resistance and high elasticity [5].

Selective laser melting (SLM) is a widely concerned technique for AM Al parts because of its high dimensional accuracy and stability [6]. However, the inherent poor flowability, high reflectivity along with high thermal conductivity make Al powders difficult to process [7,8]. The early adopters of SLM Al alloys are those designed for the traditional manufacturing processes. Among them, the near eutectic AlSi10Mg (EN AC-43000) and the eutectic AlSi12 (EN AC-44200) alloy show better process-ability by SLM, but their application is

limited by their modest mechanical properties [9–11]. Traditional precipitation-hardenable wrought Al alloys, especially 7075 alloy, have received wide attention [12], but their large solidification ranges make them more prone to hot cracking during SLM processing [13]. The crack problem has been solved by introducing nanoparticles of nucleants [13]. At the same time, by directly changing the composition of alloy powder through microalloying, high-performance Al alloys have been developed specifically for AM, including Sc and Zr microalloying Al-Mg, Al-Mn and Al-Zn-Mg-Cu alloy [14–16]. Primary $Al_3(Sc_x, Zr_{1-X})$ phases coherent with the Al matrix can act as the heterogeneous nucleation sites to refine grains, and the nano-sized $Al_3(Sc_x, Zr_{1-X})$ precipitates formed during aging can provide significant precipitation strengthening [17].

For the large-scale application of high-strength alloys, researchers have paid attention to Er, which can also form $L1_2$-structured $Al_3X$ trialuminide intermetallic compounds similar to $Al_3(Sc_x, Zr_{1-X})$ [18,19]. Recently, Guo et al. [20] investigated the strengthening mechanisms of Er-Zr composite microalloying Al-Mg alloys fabricated by laser powder bed fusion. Due to microalloying, the microstructure of those AM Al-Mg alloys is characterized by a bimodal grain structure. Equiaxed grains and epitaxial columnar grains are alternately distributed on the XZ plane, while the XY plane consists of equiaxed grains and cross-sections of the columnar grains [21]. Even so, there is no significant anisotropy in the stress–strain behavior of different building orientations [22], but the corrosion resistance of XY and XZ planes seems to be different. Gu et al. [23] studied the corrosion behavior on different planes of SLM-produced Al-4.2Mg-0.4Sc-0.2Zr alloy and found its anisotropy was related to the quite different microstructures. In order to develop high-performance AM Al alloys, it is necessary to study the relationship between microstructure and corrosion properties of Sc, Er and Zr microalloying SLMed Al alloys because excellent corrosion resistance is appealing in many applications.

In this work, dense Al-Mg-Mn-Sc-Er-Zr alloy was obtained by optimizing the process parameters of SLM. The anisotropic corrosion behaviors of the specimen fabricated with optimal parameters were performed by intergranular corrosion (IGC) tests and electrochemical measurements, including open circuit potential (OCP), potentiodynamic polarization and electrochemical impedance spectroscopy (EIS). The corresponding corrosion mechanism was analyzed based on the microstructure.

## 2. Materials and Methods

### 2.1. Powder Materials and SLM Process

The pre-alloyed powder herein was prepared by the vacuum inert gas atomization (VIGA) technique, and its morphology is shown in Figure 1c. It has a nominal composition of Al-4.74Mg-0.52Mn-0.41Sc-0.36Er-0.43Zr (wt.%), which was detected by inductively coupled plasma atomic emission spectroscopy (ICP-AES, 725, Agilent, Palo Alto, CA, USA). Powders for the SLM process were sieved in the range of 15–53 μm, and their particle size was measured by a particle size analyzer (LS 13 320, Beckman Coulter, Bria, CA, USA).

Samples were fabricated using an SLM machine (ASA-260M, CASIC, Beijing, China) outfitted with a 500 W fiber-laser, and Figure 1a depicts the working principle of the system. The scanning strategy of the laser was a stripe hatch strategy with a 67° rotation in adjacent layers (Figure 1b). Cubic samples ($10 \times 10 \times 10$ mm$^3$) were fabricated to determine the optimal process parameters which are based on the relative density results and metallographic analysis. Cubic samples ($10 \times 10 \times 10$ mm$^3$) were fabricated for microstructure characterization and electrochemical tests. Cuboid samples ($40 \times 25 \times 8$ mm$^3$) were manufactured for IGC tests.

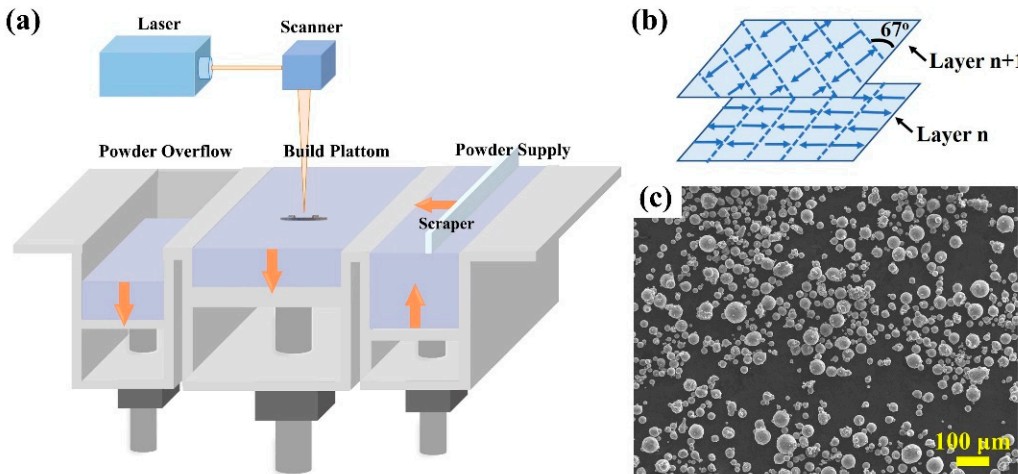

**Figure 1.** (**a**) Schematic diagram of SLM; (**b**) scanning strategy; (**c**) powder morphology.

The designed parameters for process optimization were laser power ($P$) of 340–400 W, scanning speed ($v$) of 500–1100 mm·s$^{-1}$, hatch spacing ($h$) of 0.11 mm and layer thickness ($t$) of 0.03 mm. The density test is a convenient way to evaluate the processing quality of AM part [24]. The actual density of the sample was measured by Archimedes method, and the theoretical density was 2.676 g·cm$^{-3}$. The volume energy density ($ED_v$) of laser is commonly used to optimize process parameters of SLM, which is calculated as follows [6]:

$$ED_v = P/vht, \tag{1}$$

where $P$ (W), $v$ (mm·s$^{-1}$), $h$ (mm) and $t$ (mm) denote the laser powder, scan speed, hatch spacing and layer thickness, respectively.

### 2.2. Measurements of Corrosion Behavior

IGC tests were carried out in 1000 mL deionized water containing 30 g NaCl and 10 mL HCl for 24 h (25 °C). Before corrosion, the polished sample was cleaned with alcohol and immersed in NaOH solution, then washed with deionized water and HNO$_3$ solution successively until the surface was smooth. An electrochemical workstation (PARSTAT 3000A, AMETEK, San Diego, CA, USA) was applied to obtain the electrochemical corrosion properties. The corrosion process was operated with a sealed traditional three-electrode system in 3.5 wt.% NaCl solution at 25 °C. The sample acted as the working electrode with an exposed area of 1 cm$^2$. The testing surface was mechanically ground to 7000-grit and polished. A platinum sheet and a saturated calomel electrode (SCE) with a salt bridge served as the counter electrode and the reference electrode, respectively. Prior to the test, the working electrode was placed in the electrolyte for 1 h to obtain a stable OCP. EIS tests were carried out under the OCP, and the frequency ranged from 100 kHz to 10 mHz with an amplitude of 10 mV. The potentiodynamic polarization test was scanned from −1.7 to −0.4 V with a scanning rate of 2 mV·s$^{-1}$. For the electrochemical tests, each plane was tested three times to ensure data reproducibility and that the average curves are shown.

### 2.3. Microstructure Characterizations

Metallographic sections of the as-built sample on different planes were observed by optical microscopy (OM, Axio Vert. A1, Zeiss, Oberkochen, Germany). A field emission scanning electron microscope (FE-SEM, JEOL JSM-IT700HR, Musashino, Japan) was employed to characterize the microstructure and corrosion morphologies of as-built samples on different planes. The surface morphology contour after IGC tests was measured by an ultra-depth 3D microscope (VHX-7000, Keyence, Osaka, Japan). Phase identification was analyzed by an X-ray diffractometer (XRD, SmartLab, Rigaku, Akishima, Japan) with Cu Kα radiation. The 2θ angle ranged from 20° to 90° using the scanning speed of 4°·min$^{-1}$.

Features of grains and grain boundaries were characterized by electron backscattered diffraction (EBSD). EBSD specimens were prepared by electrolytic polishing (electrolyte: 90 vol.% ethanol + 10 vol.% perchloric acid). Characterization of primary phases was carried out on a transmission electron microscope (TEM, F200X S/TEM, Talos, Thermo Scientific, Waltham, MA, USA). The TEM foils were prepared by mechanical grinding and twin-jet polishing (electrolyte: 30 vol.% $HNO_3$ + 70 vol.% methanol).

## 3. Results

### 3.1. Densification

Figure 2a describes the densification of samples with different process parameters, indicating that the relative density of the sample increases first and then decreases as the volume energy density increases. The densest part (99.3%) was obtained when the $ED_v$ was 127.95 J·mm$^{-3}$. In Figure 2b, when the laser power exceeds 380 W, a low scanning speed reduces the relative density of the samples. For the same laser power, the sample fabricated at the scanning speed of 900 mm·s$^{-1}$ had the highest relative density.

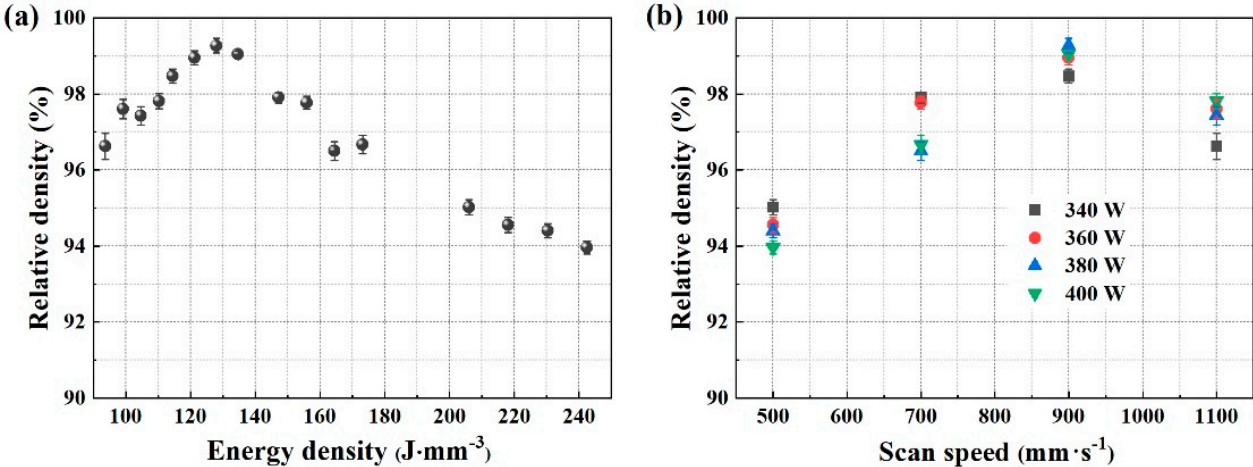

**Figure 2.** Densification of cubic samples with different process parameters: (**a**) volume energy density; (**b**) laser power and scanning speed.

Metallographic images and the size distribution of pores of as-built samples with different process parameters are shown in Figure 3. No cracks were found in those parts, so the objective of parameter optimization for SLM Al is minimizing porosity in the fabricated parts. When the lower values of scanning speed (500 mm·s$^{-1}$) are applied, some spherical pores of tens of microns appear, indicating the "keyhole mode" laser melting. There are also small pores of less than 10 microns, and their formation is attributed to entrapped shielding gas or the evaporation of elements [25]. As the scanning speed increases to 1100 mm·s$^{-1}$, the porosity of the sample increases. This is related to the non-molten powder and the undamaged oxide film outside the semi-solid melt, which are difficult to be remelted by the molten pool of subsequent layers due to the low laser $ED_v$ [26]. The specimen with minimal porosity is obtained as $P$ is 380 W and $v$ is 900 mm·s$^{-1}$, which are used as the optimal parameters to process samples in the following research.

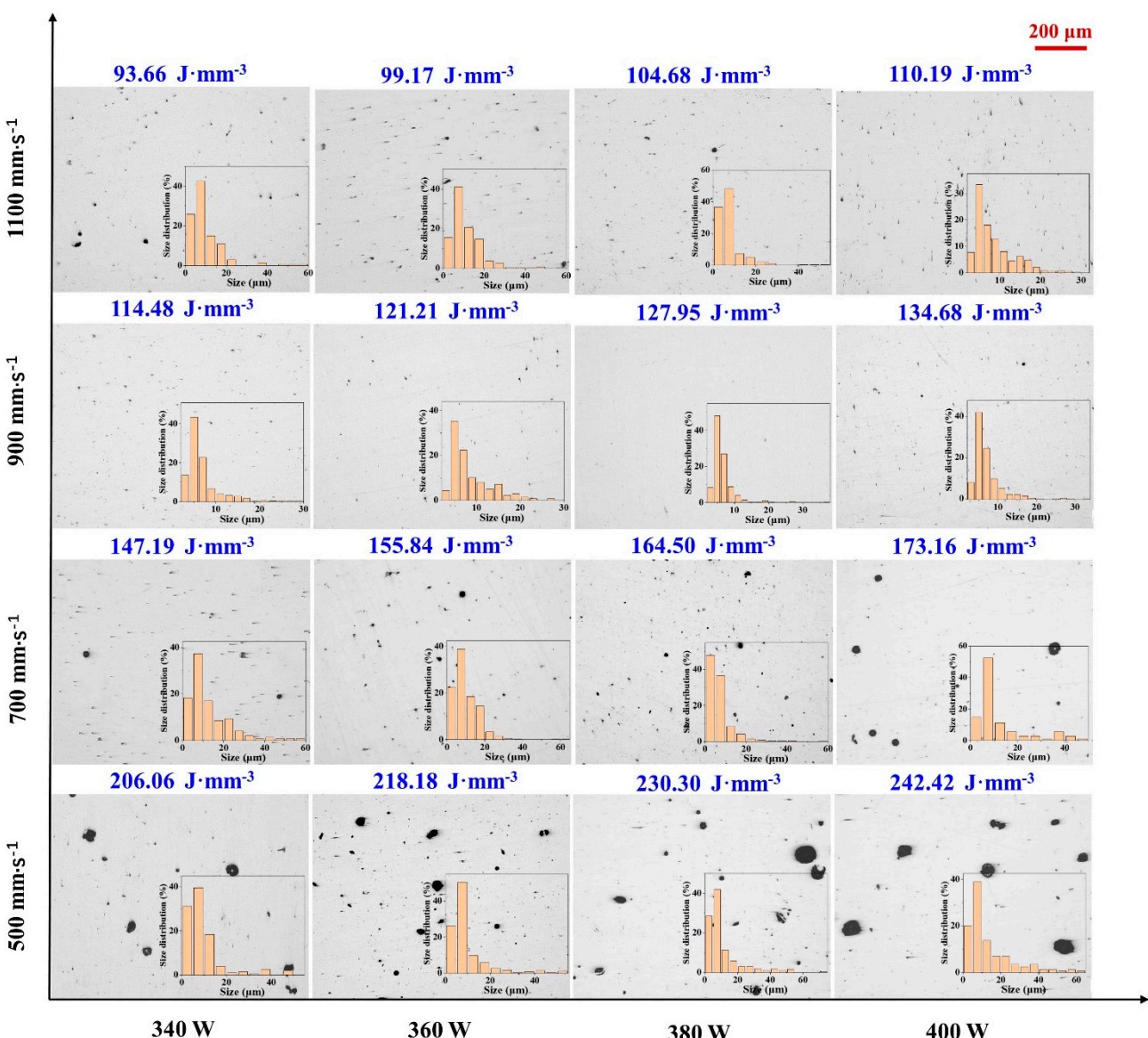

**Figure 3.** The morphology and the size distribution of pores of as-built samples with different process parameters.

*3.2. Microstructure*

The layer-wise building of SLM results in a dynamic solidification condition, so the microstructure along the building direction (XZ plane) and perpendicular to the building direction (XY plane) are different. The XY plane presents a stack of individual scan tracks overlapped horizontally (Figure 4a), while the XZ plane is a cross-section of overlapped molten pools, showing a fish-scale pattern (Figure 4b). Furthermore, the amplified microstructures of each plane are obtained by backscattered SEM images, as Figure 4c,d shows. Similar to Zr-modified [27], Sc- and Zr-modified [14], Er- and Zr-modified [20] Al alloys, the microstructure of this Sc-, Er- and Zr-modified Al alloy consists of equiaxed grains (less than 1 μm) located at the melt pool boundary and columnar grains towards the top-center of the melt pool. The equiaxed grain with a larger diameter in Figure 4c is the cross-section of columnar grains. There is stress at the solid–liquid interface during the solidification process, and the stress concentration is highest at the intersection of columnar dendrites with different growth directions [28]. The equiaxed, fine grain can accommodate strain in the semi-solid state and lock the orientation of dendrites, so this bi-modal grain structure

can effectively inhibit the initiation of cracks [13]. In addition, a number of primary phases were generated in the fine grains (FG) region but less in the columnar grains (CG) region.

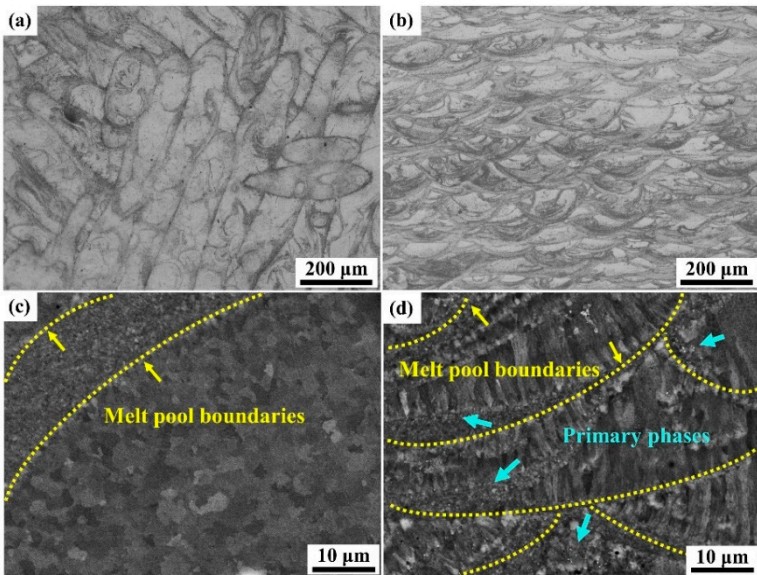

**Figure 4.** Micrograph of as-built sample on XY plane: (**a**) metallographic structure and (**b**) backscatter SEM; XZ plane: (**c**) metallographic structure and (**d**) backscatter SEM.

EBSD results of as-built samples on different planes are shown in Figure 5. From the inverse pole figure (IPF) and the pole figure (PF) in Figure 5a,b,d,e, it is worth noting that FG regions comprise equiaxed grains with a random crystallographic orientation, while the columnar grains of CG regions reveal a strong <001> texture. Furthermore, as shown in Figure 5c,f, the XZ plane contains more sub-micron equiaxed grains compared with the XY plane for the given visual field, so the texture intensity of the XY plane in the [001] direction (building direction) is higher. The heterogeneous nucleation and the preferred solidification direction of Al result in this bi-modal grain structure, in which the above-mentioned primary phases act as nucleation sites to form FG regions, and the thermal gradient provides the driving force for the epitaxial growth of columnar grains [15,27].

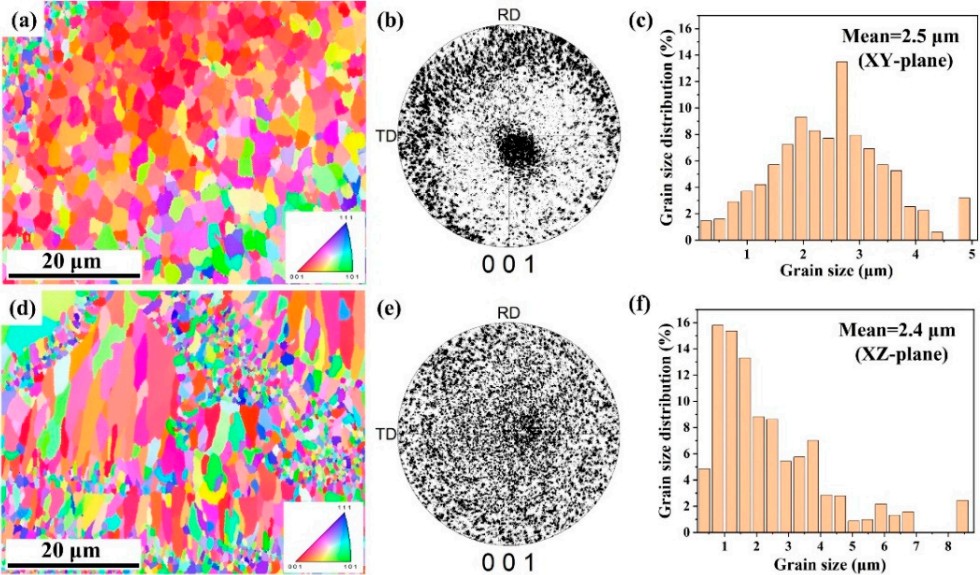

**Figure 5.** The EBSD results of the as−built sample on XY plane: (**a**) IPF, (**b**) [001] PF and (**c**) grain size distribution; XZ plane: (**d**) IPF; (**e**) [001] PF; (**f**) grain size distribution.

Grain boundary maps of Figure 5a,d are exhibited in Figure 6a,c, respectively, and the corresponding quantitative analysis results are presented in Figure 6b,d. The total length of the grain boundary is 5.67 mm for the XY plane, 6.13 mm for the XZ plane, and the number of grains is 98,249 for the XY plane and 106,106 for X, so the total length of the grain boundary and the number of grains per unit area shows little difference on different planes. In addition, it can be observed that the fraction of low-angle grain boundaries (LAGBs) ($2° \leq \theta \leq 15°$) on the XY plane (29.7%) is higher than that on the XZ plane (11.8%).

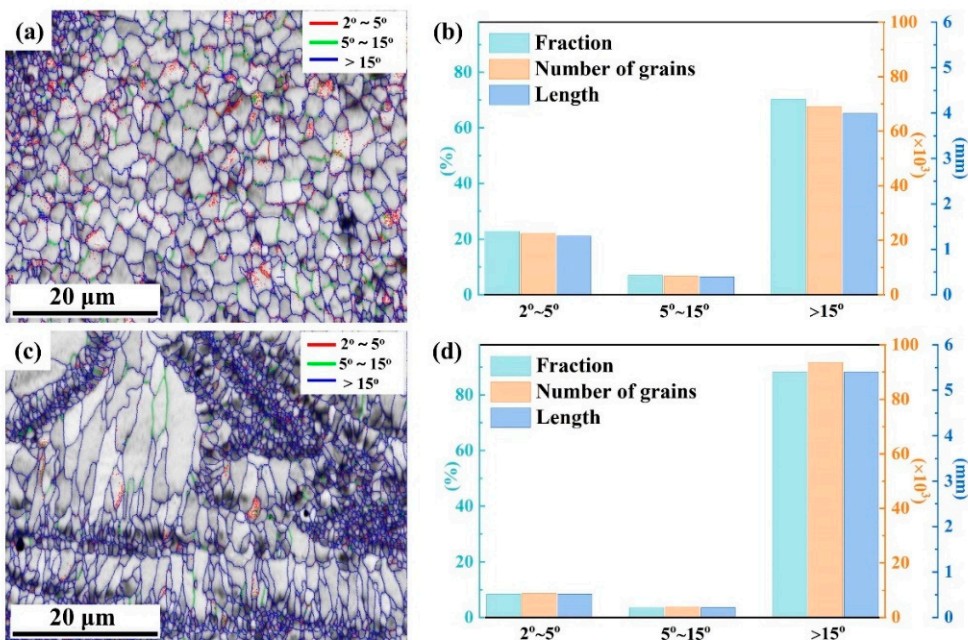

**Figure 6.** Grain boundary maps of the as-built sample: (**a**) XY plane; (**b**) quantitative results of (**a**); (**c**) XZ plane; (**d**) quantitative results of (**c**).

### 3.3. Phase Characterization

Figure 7 shows the XRD patterns of the as-built sample on different planes, in which Bragg diffraction peaks of the Al matrix are marked according to the standard XRD spectrum. For the XY plane, the abnormally elevated intensity of the (200) diffraction peak also confirms the existence of a strong {200} fiber texture. Given that the phase identification of XRD is limited by resolution, primary phases were further investigated via TEM examinations.

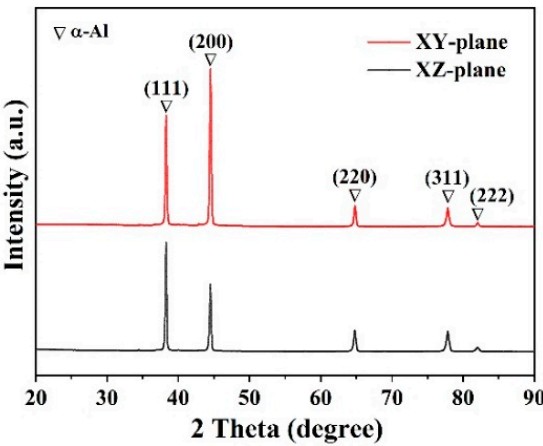

**Figure 7.** XRD patterns of as-built sample for different planes.

A bright field (BF)-STEM image taken from the FG region is presented in Figure 8a, and a bean-like phase within a grain is indicated by a red dotted box. Based on the energy dispersive X-ray spectroscopy (EDX) analysis (Figure 8b) and the selected area electron diffraction (SAED) pattern along the [011] zone axis (Figure 8c), the phase is identified as L1$_2$-structured Al$_3$(Sc, Er, Zr). The high resolution (HR)-TEM image of this particle (Figure 8d) reveals a cube-on-cube orientation relationship with the Al matrix, so it is difficult to recognize the primary phase from XRD. Al$_3$(Sc, Er, Zr), a non-stoichiometric compound with a core/double-shell structure, can be an ideal seed for fcc-Al growth as the similar lattice constant with α-Al (4.049 Å) [12,29,30]. Due to the high temperature in the central area at the top of the molten pool, primary Al$_3$(Sc, Er, Zr) particles mainly exist in FG regions during the heat cycle of the SLM process [14].

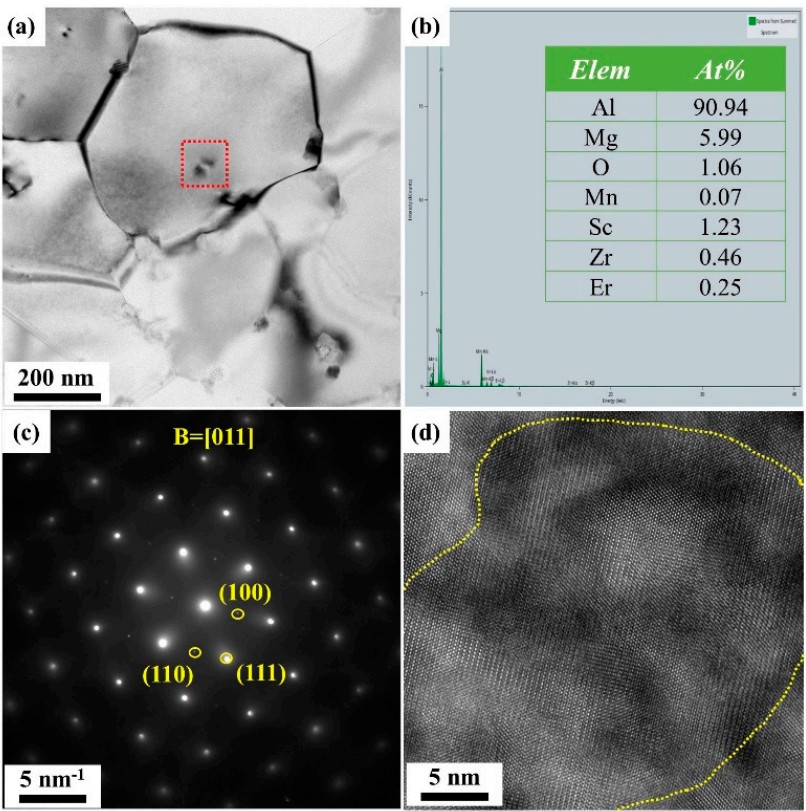

**Figure 8.** TEM images of primary Al$_3$(Sc, Er, Zr) in as-built sample: (**a**) BF-STEM image of the FG region; (**b**) EDX results; (**c**) SAED; (**d**) HR-TEM.

Al-Mg-oxide particles are a common primary phase in the sample, which has been reported by Ma et al. [31] and Spierings et al. [14]. The oxide inclusions created during the laser melting process and the oxide film of powders can provide sufficient oxygen. Figure 9a depicts the DF-TEM image of the FG region. The EDX results (Figure 9b) and the SAED pattern (Figure 9c) signify that the white rhombic particle less than 50 nm long is the Al$_2$MgO$_4$ phase with spinel structure [32]. This particle can promote the nucleation of the Al$_3$(Sc, Er, Zr) precipitation and then the mixed phases can act as a seed for fcc-Al growth to form fine-grain regions. With a high melting point of 2408 K, some intragranular and intergranular Al$_2$MgO$_4$ particles can survive at CG regions, as Figure 9d shows. Furthermore, STEM-EDX elemental mapping indicated that there is no other primary phase in the CG region because the rapid solidification of SLM increases the solid solubility of elements in the matrix.

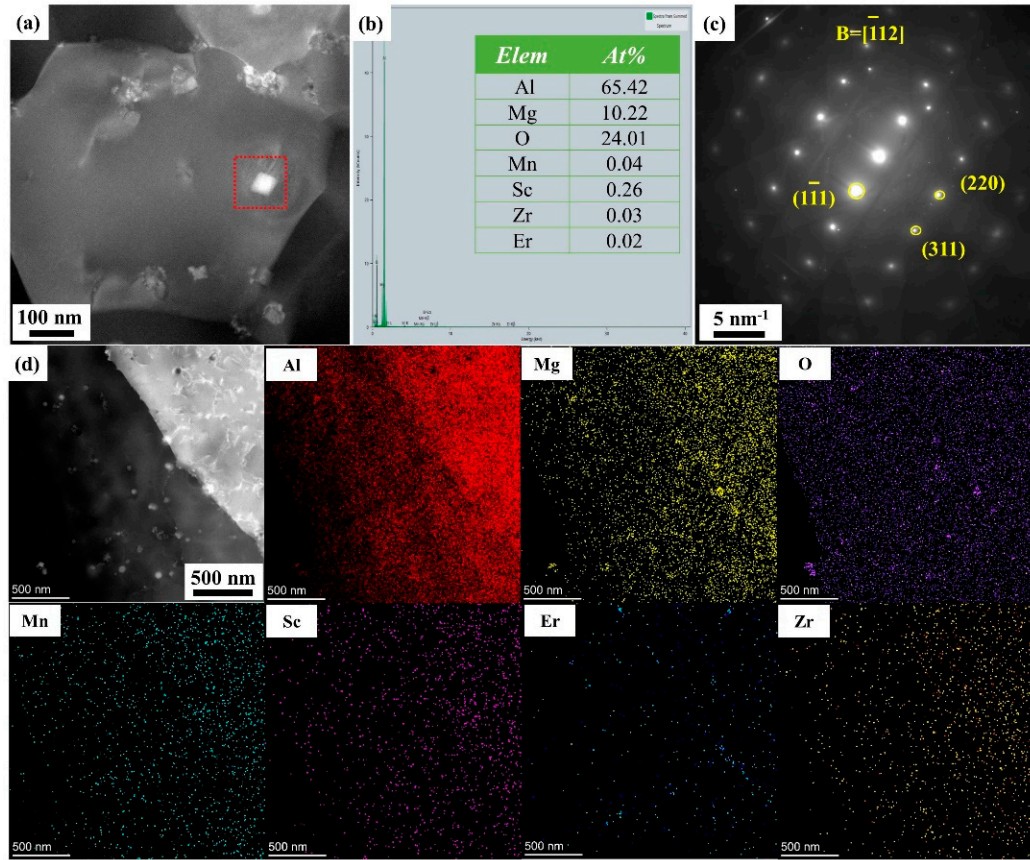

**Figure 9.** TEM images of Al$_2$MgO$_4$ in as-built sample: (**a**) DF-STEM image of the FG region; (**b**) EDX results; (**c**) SAED; (**d**) BF-STEM image of the CG region and the corresponding STEM-EDX mapping analysis of Al, Mg, O, Mn, Sc, Er and Zr elements.

### 3.4. Corrosion Behavior

Corrosion of the Al alloy begins with the interaction between the surface oxide film and aggressive anions (such as Cl$^-$) [33]. Once the integrity of the passivated film is destroyed, corrosion can easily occur along grain boundaries. However, it is difficult to directly measure the corroded depth of grain boundaries from the cross-sectional micrographs after the IGC test as the grain size of SLMed samples is too small. Gu et al. [34] used the maximum corrosion depth of the longitudinal section to characterize the IGC level of SLMed Al-Mg-Sc-Zr alloy. Here, the IGC behavior of samples on different planes was characterized by 3D surface morphologies of corroded samples and the internal height of pits. In Figure 10a,b, the XZ plane is coarser compared with the XY plane, indicating an inferior IGC resistance of the XZ plane. For their corresponding actual microscope morphologies in Figure 10c,d, the black area is a deep pit, and there are more deep pits on the XZ plane. Furthermore, the quantitative analysis of the pit shows that the corroded pit of the XZ plane is deeper and larger.

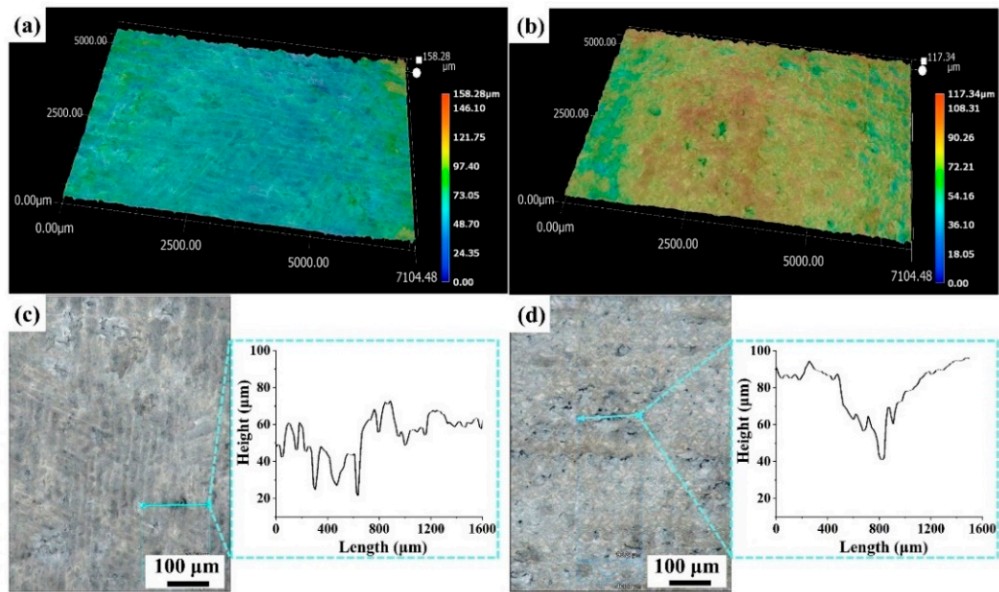

**Figure 10.** Three-dimensional surface morphologies of the as-built alloy after IGC tests: (**a**) surface roughness of XY plane; (**b**) surface roughness of XZ plane; (**c**) corrosion morphology and the depth of a corrosion pit for (**a**); (**d**) corrosion morphology and the depth of a corrosion pit for (**b**).

The open circuit potential (OCP) evolution with time for the as-built samples tested in 3.5 wt.% NaCl solution (25 °C) was monitored prior to performing other electrochemical tests, and the results are shown in Figure 11a. The OCP values of the XY and XZ planes are stabilized at −901.54 mV (vs. SCE) and −989.05 mV (vs. SCE), respectively, indicating that the XY plane shows better corrosion resistance. Figure 11b presents the potentiodynamic polarization curves with a significant passivation area. Values of corrosion current density ($I_{corr}$), corrosion potential ($E_{corr}$), anodic Tafel slope ($\beta_a$) and cathodic Tafel slope ($\beta_c$) can be obtained by linear fitting, and relevant corrosion parameter values are listed in Table 1. The polarization resistance ($R_p$) was calculated by [35]:

$$R_P = \beta_a \beta_c / [2.3 \cdot (\beta_a + \beta_c) \times I_{corr}]. \tag{2}$$

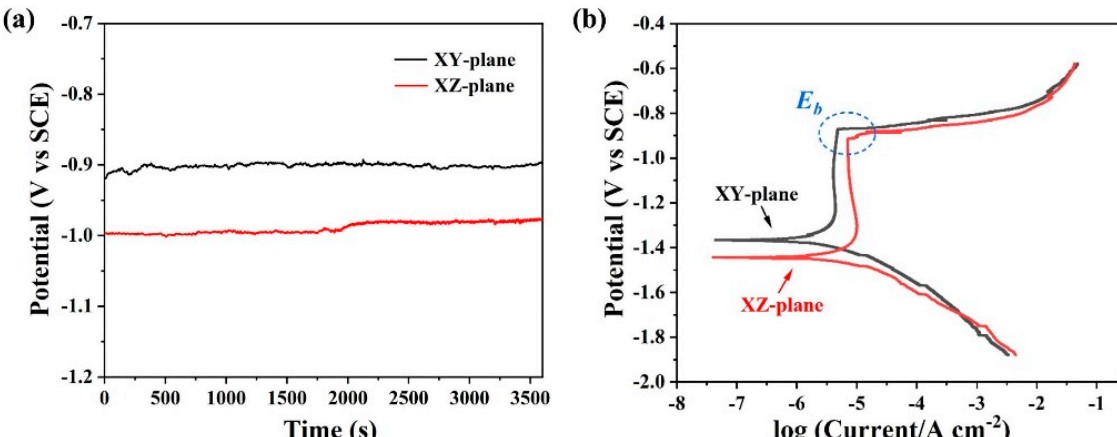

**Figure 11.** Electrochemical measurement results of samples for different planes in 3.5 wt.% NaCl solution (25 °C): (**a**) OCP curves and (**b**) potentiodynamic polarization curves.

**Table 1.** Fitting results of polarization curves for as-built samples on different planes.

| Samples | - | $I_{corr}$ (µA·cm$^{-2}$) | $R_p$ (Ω·cm$^2$) | $E_{corr}$ (mV) | $E_b$ (mV) |
|---|---|---|---|---|---|
| XY plane | Average | 2.512 | $2.999 \times 10^4$ | −1354.83 | −883.31 |
| | Std-Dev | 0.837 | $2.759 \times 10^3$ | 21.72 | 9.46 |
| XZ plane | Average | 4.681 | $1.972 \times 10^4$ | −1455.47 | −928.45 |
| | Std-Dev | 1.110 | $2.974 \times 10^3$ | 9.66 | 27.33 |

The corrosion current is directly proportional to the electrochemical corrosion rate, so a low corrosion current density coupled with a large polarization resistance indicates an increase in the corrosion resistance of the material [36]. From the results, the XY plane has better corrosion resistance since it has a lower $I_{corr}$ (2.512 µA·cm$^{-2}$), a higher $E_{corr}$ (−1354.83 mV) and a higher $R_p$ ($2.999 \times 10^4$ Ω·cm$^2$) compared to the XZ plane ($I_{corr}$ = 4.681 µA·cm$^{-2}$; $E_{corr}$ = −1.455.47 mV; $R_p$ = $1.972 \times 10^4$ Ω·cm$^2$). Furthermore, the XY plane has a higher breakdown potential ($E_b$), which means that the stability of the passive film on the surface is better than the XZ plane.

In order to quantitatively evaluate the barrier properties of the surface oxide film of samples, EIS measurements were performed. Based on the fact that Nyquist plots (Figure 12a) consist of two capacitive loops deviating from the semicircle, the alloy–solution interface has an electric double-layer structure, and its equivalent circuit is shown in Figure 12b. $R_s$ is the resistance of the electrolyte. $R_f$ and $CPE_1$ correspond to the polarization resistance and capacitance of the passive film, respectively. $R_{ct}$ is Faradic charge transfer resistance, and $CPE_2$ describes its resistance. Thus, it can be seen that there are two peaks in the phase angle bode plot (Figure 12c). The peak at approximately 80° reveals the formation of oxide film on the surface, and another one is related to the blocking effect of Al(OH)$_3$ on Cl$^-$ migration [37]. In Figure 12d, the impedance moduli with a wide linear part at middle frequency reveal that the property of oxide film dominates the impedance behavior. The capacitance can be described by film thickness through the expression [38]:

$$C = \varepsilon \varepsilon_0 A / d, \tag{3}$$

where $\varepsilon$ is the dielectric constant, $\varepsilon_0$ is the permittivity of vacuum and A and $d$ represent the surface area and thickness of the film, respectively. Furthermore, the effective capacitance related to the CPE is defined as [39]:

$$C_{\text{eff}} = [Q \times R_f^{(1 - \alpha)}]^{-\alpha}, \tag{4}$$

where Q and $\alpha$ are CPE parameters and $R_f$ is the film resistance on the surface, so $d$ is inversely proportional to $[Q \times R_f^{(1 - \alpha)}]^{-\alpha}$. According to the fitting results of the parameters in Table 2 and Equations (1) and (2), it can be determined that the $d$ of the XY plane is thicker. Moreover, the $R_f$ of the XY plane (67.22 kΩ·cm$^2$) is higher than that of the XZ plane (51.78 kΩ·cm$^2$). Therefore, the XY plane of the SLMed sample possesses a superior corrosion resistance, which is consistent with the conclusion of the potentiodynamic polarization tests.

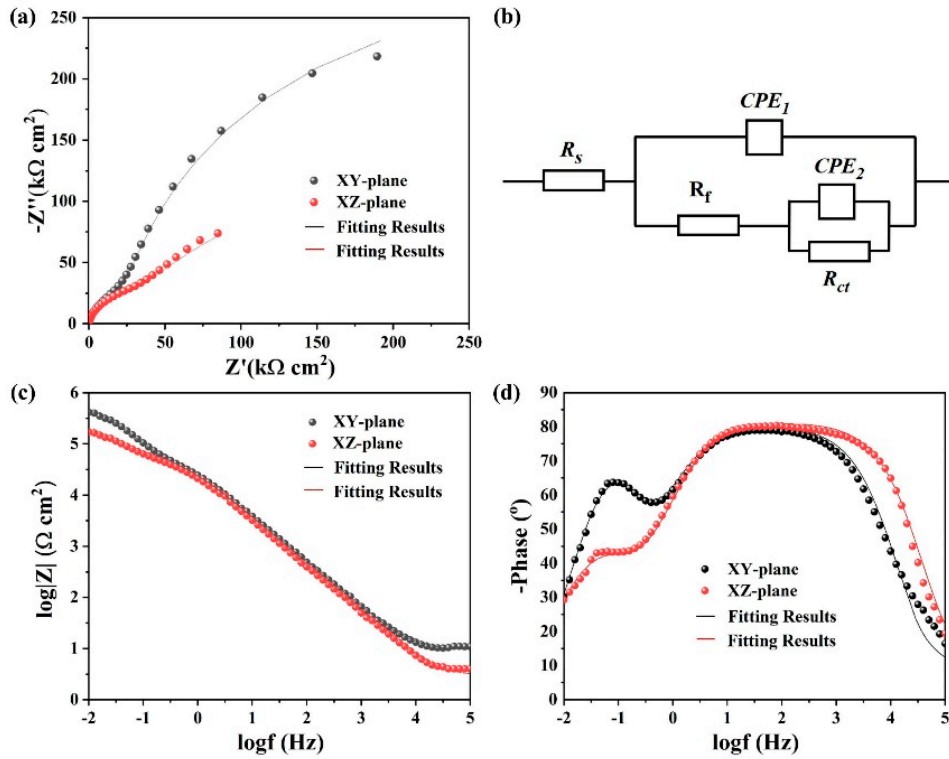

**Figure 12.** EIS results of samples for different planes in 3.5 wt.% NaCl solution (25 °C): (**a**) Nyquist plots; (**b**) equivalent electrical circuit for the EIS test; (**c**,**d**) bode plots.

**Table 2.** Fitting results of EIS measurements for as-built samples on different planes.

| Samples | - | $R_s$ ($\Omega \cdot cm^2$) | $R_f$ ($k\Omega \cdot cm^2$) | $CPE_1$ ($F \cdot cm^{-2}$) | $n_1$ | $R_{ct}$ ($k\Omega \cdot cm^2$) | $CPE_2$ ($F \cdot cm^{-2}$) | $n_2$ |
|---------|---|------|------|------|------|------|------|------|
| XY plane | Average | 8.89 | 67.22 | $6.67 \times 10^{-6}$ | 0.88 | 476.63 | $1.08 \times 10^{-5}$ | 0.97 |
|          | Std-Dev | 0.25 | 3.52 | $7.80 \times 10^{-8}$ | 0.01 | 18.50 | $3.65 \times 10^{-7}$ | 0.02 |
| XZ plane | Average | 2.88 | 51.78 | $7.55 \times 10^{-6}$ | 0.89 | 253.23 | $2.92 \times 10^{-5}$ | 0.76 |
|          | Std-Dev | 0.34 | 1.57 | $2.24 \times 10^{-7}$ | 0.01 | 20.58 | $1.04 \times 10^{-6}$ | 0.02 |

## 4. Discussion

Pores are a common defect in SLMed alloys. Zhang et al. [40] studied the anisotropic corrosion resistance of Ni-based composites fabricated by selective laser melting and confirmed that pores affected the passivity of the metals and are preferential sites for corrosion inducing in solutions. Therefore, in this study, after densifying the SLM-processed sample, the phases and characteristics of grains were investigated to explain the anisotropic corrosion mechanism.

By microalloying Sc, Er and Zr elements, FG regions composed of submicron grains in Figure 5 can accommodate strain in the semi-solid state by suppressing coherency, thus obtaining the crack-free microstructure [12]. Meanwhile, the primary $Al_3$(Sc, Er, Zr) phase (Figure 8) and the $Al_2MgO_4$ phase (Figure 9) are not only nucleating agents of FG regions but also affect the property of oxide films. It has been confirmed that the Sc containing intermetallic compounds with ordered $L1_2$ structure act as the cathodic to promote the initiation of localized corrosion [34,41,42]. Although the corrosion behavior of the $Al_2MgO_4$ phase has not been mentioned in previous research on SLMed Al-Mg alloys [22,24,34], the precipitates may accelerate micro-galvanic corrosion due to their differences in potentials [43]. As Figure 4 shows, the primary phases (white particles) of an as-built alloy mainly exist in FG regions, and only a small amount of the $Al_2MgO_4$

phase is distributed in CG regions. Furthermore, the STEM-EDX mapping analysis of CG regions (Figure 9d) reveals that Mn, Sc, Er and Zr elements are retained in solid solution without forming intermetallic compounds, and no other Al-Mg phase is found on the grain boundary as well. This is because the high cooling rate ($\sim10^5$–$10^6$ K·s$^{-1}$) during the SLM process improves the solid solubility of elements, and the temperature (up to $\sim2000$ °C) in CG regions exceeds the melting point of most phases. From the grain size distribution of Figure 5, the proportion of FG regions in the XZ plane is higher than the XY plane, so the XZ plane possesses inferior pitting corrosion resistance induced by phases.

Zhang et al. [44] concluded that the anisotropic corrosion behavior of laser powder bed fusion Al-Mn-Mg-Sc-Zr alloy was related to the difference in grain size and local corrosion induced by phases. Our research came to a similar conclusion and also found that the texture strength and the orientation difference of adjacent grains can affect the corrosion anisotropy of the sample. Ralston et al. [45] revealed that the relationship between the grain size and corrosion rate of Al can be considered analogous to the bounds of the Hall–Petch relationship, so the corrosion resistance improves with the decrease in grain size. Statistical results of Figure 5 show that the average grain size of the XY plane (2.47 µm) and the XZ plane (2.44 µm) have no significant difference. As a result, the grain size has little contribution to the anisotropic corrosion behavior. The differences between the microstructure of the matrix and the chemical composition of the grain boundaries can lead to corrosion, and the phases in high-angle grain boundaries (HAGBs) are inclined to preferentially precipitate and aggregate [46]. In Figure 6c,d, the fraction of LAGBs on the XY plane (29.7%) is more than twice that on the XZ plane (11.8%); hence, the XZ plane of samples is prone to corrosion compared with the XY plane. In terms of crystallographic orientation, grains with the (001) preferred orientation are the most resistant to pitting corrosion for Al alloy [47]. The susceptibility to pitting corrosion in chloride solution is generally attributed to the surface energy of different planes, and Schochlin et al. [48] obtained the surface energy of Al alloy by first-principles total-energy calculation in the order (100) > (110) $\approx$ (111). Figure 5c,d show that both the XY plane and XZ plane present a (001) texture along the building direction, but the XY plane has a stronger texture index, resulting in better pitting corrosion resistance of the XY plane.

## 5. Conclusions

In this work, the Al-Mg-Mn-Sc-Er-Zr alloy was produced by selective laser melting. The densification, microstructure and anisotropic corrosion behavior were investigated. The main conclusions can be drawn as follows:

Dense SLMed samples were obtained by microalloying and optimization of process parameters, which is indispensable for the study of corrosion anisotropy. The SLMed Al-Mg-Mn-Sc-Er-Zr alloy exhibited a bi-modal grain and crack-free structure. The relative density of as-built samples first increased and decreased with the increase in volume energy density. The densest sample (99.3%) was obtained at ED$_v$ of 127.95 J·mm$^{-3}$.

The XY plane of the as-built sample was less prone to IGC. Electrochemical measurements also indicate that the XY plane possesses better corrosion resistance because it has a higher OCP value of $-901.54$ mV, a higher $R_p$ of $2.999 \times 10^4$ Ω·cm$^2$, a lower $I_{corr}$ of 2.512 µA·cm$^{-2}$ as well as the passive film with superior corrosion resistance.

Al$_3$(Sc, Er, Zr) and Al$_2$MgO$_4$ phases of an as-built alloy not only act as nuclei for heterogeneous nucleation but also promote the initiation of localized corrosion. With a higher proportion of FG regions, the XZ plane possesses inferior pitting corrosion resistance compared with the XY plane.

The influence of grain size on anisotropic corrosion behavior is small because there is no significant difference in grain size between the XY plane (2.47 µm) and the XZ plane (2.44 µm). The XY plane (29.7%) has a higher fraction of LAGBs than the XZ plane (11.8%), and the (001) texture index of the XY plane is stronger along the building direction, which contributes to the better corrosion resistance of the XY plane.

**Author Contributions:** Conceptualization, J.S. and Q.H.; methodology, J.S. and J.L.; software, W.X. and J.Z.; validation, X.Z. and Y.C.; formal analysis, X.Z. and J.S.; investigation, J.L. and Y.C.; resources, Q.H.; writing—original draft preparation, J.S.; writing—review and editing, J.S.; supervision, Q.H. All authors have read and agreed to the published version of the manuscript.

**Funding:** This work was funded by the National Key R & D Program of China [2021YFB3701201].

**Institutional Review Board Statement:** Not applicable.

**Informed Consent Statement:** Not applicable.

**Data Availability Statement:** Not applicable.

**Acknowledgments:** This work was financially supported by the National Key R & D Program of China [2021YFB3701201].

**Conflicts of Interest:** The authors declare that they have no known competing financial interest or personal relationships that could have appeared to influence the work reported in this paper.

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
