# Peer review of "Densification, Microstructure and Anisotropic Corrosion Behavior of Al-Mg-Mn-Sc-Er-Zr Alloy Processed by Selective Laser Melting"

_coatings, doi:10.3390/coatings13020337_

Round 1
Reviewer 1 Report
The work presents the results of testing an aluminum alloy produced by the SLM method. Structural and electrochemical tests were performed. The work lacks references to the results obtained by other authors, and thus a discussion of the results obtained. The comments below also require clarification and supplementation:
1) The method of determining the density of the material, as well as the method of preparing samples for its assessment, require a description.
2) Figure 3 - axes need description. As shown in the Figure, the porosity decreases as the scan speed increases. At 1100 mm*s-1, the porosity increases again. This requires a comment..
3) Please mark (contour) the identified phase in the HR-TEM image (Fig. 8d). It is illegible in its present form. Was this the only phase identified besides the matrix? The presence of magnesium should allow the formation of different phases.
4) On how many samples were electrochemical tests carried out? Are the curves shown random or average?
5) Has crystallographic attack corrosion been observed for these alloys? On what basis is the XY plane considered to have better corrosion resistance? What was the reason?
A different font color is visible in the Conclusions. Please correct it.
Reviewer 2 Report
The manuscript "Densification, microstructure and anisotropic corrosion behavior of Al-Mg-Mn-Sc-Er-Zr alloy processed by selective laser melting" deals with some interesting aspects of selective laser melting of aluminium alloys. This is a very challenging issue due to the nature of aluminium alloys. The authors have represented a very nice work with interesting data. I recommend this manuscript for publication after some mandatory revisions:
1. Though the manuscript is about precipitation-hardenable aluminium alloys, the literature on these alloys is rather poor. Authors are recommended to present a more comprehensive introduction on aluminium alloys and their properties. I suggest authors consult following manuscripts in their revision:
- Effect of aging plus cryogenic treatment on the machinability of 7075 aluminum alloy. Vacuum, 208, 2023, 111692. doi: https://doi.org/10.1016/j.vacuum.2022.111692
- Effect of heat treatment process on the micro machinability of 7075 aluminum alloy. Vacuum, 207, 2023, 111574. doi: https://doi.org/10.1016/j.vacuum.2022.111574
2. Authors have mentioned that their process has ended up in crack-free samples. As cracking is a critical issue in SLMed aluminium alloys, this needs to be further elaborated. What are mechanisms of crack formation in laser/powder interactions and how this is optimized in your work need to be (briefly) discussed. The following paper is highly recommended to be referred. Though it is in a different system, it is still very informative and highly recommended.
- Investigation of welding crack in micro laser welded NiTiNb shape memory alloy and Ti6Al4V alloy dissimilar metals joints. Optics & Laser Technology, 91, 2017, 197-202. doi: https://doi.org/10.1016/j.optlastec.2016.12.028
3. Please provide quantitative data from Figure 3 (i.e. the size distribution of voids). Use image analysing techniques.
4. Hardly is there any data on the mechanical properties of samples. What is the implication of this Al-Mg-O phase for the mechanical properties of samples?
5. Please do not report grain size in two digit accuracy (i.e. 2.47 um). I suggest you report 2.5 um instead.
Round 2
Reviewer 1 Report
I have no more objections.
Reviewer 2 Report
Authors have nicely revised the manuscript. I believe this paper can be published in this journal.